# Understanding Top-k Sparsification in Distributed Deep Learning

## Abstract

Distributed stochastic gradient descent (SGD) algorithms are widely deployed in training large-scale deep learning models, while the communication overhead among workers becomes the new system bottleneck. Recently proposed gradient sparsification techniques, especially Top-$k$ sparsification with error compensation (TopK-SGD), can significantly reduce the communication traffic without obvious impact on the model accuracy. Some theoretical studies have been carried out to analyze the convergence property of TopK-SGD. However, existing studies do not dive into the details of Top-$k$ operator in gradient sparsification and use relaxed bounds (e.g., exact bound of Random-$k$) for analysis; hence the derived results cannot well describe the real convergence performance of TopK-SGD. To this end, we first study the gradient distributions of TopK-SGD during training process through extensive experiments. We then theoretically derive a tighter bound for the Top-$k$ operator. Finally, we exploit the property of gradient distribution to propose an approximate top-$k$ selection algorithm, which is computing-efficient for GPUs, to improve the scaling efficiency of TopK-SGD by significantly reducing the computing overhead.

## 1 Introduction

Training large-scale deep neural networks (DNNs) generally exploits distributed synchronous stochastic gradient descent (SGD) optimization algorithms to reduce the overall training time. Let $P$ be the number of workers in a distributed setting, and $\boldsymbol{x} \in \mathbb{R}^d$ denotes the model parameters with $d$ dimensions. At the $t$-th iteration, distributed synchronous SGD updates the model parameters by

$$\boldsymbol{x}_{t+1} = \boldsymbol{x}_t - \eta_t \frac{1}{P} \sum_{p=1}^{P} \boldsymbol{g}_t^p, \tag{1}$$

where $\boldsymbol{g}_t^p \in \mathbb{R}^d$ is the stochastic gradient with its locally selected data for the loss function $f^p(\boldsymbol{x}) : \mathbb{R}^d \to \mathbb{R}$ and $\eta_t$ is the learning rate. The aggregation of $d$-dimension gradients from $P$ workers requires a communication complexity of $O(d)$ in terms of communication traffics[1], which generally limits the system scalability. Gradient sparsification (Strom, 2015; Dryden et al., 2016; Aji & Heafield, 2017; Chen et al., 2018; Lin et al., 2018) is a promising technique for distributed SGD, which can significantly reduce the communication traffic while reserving the model convergence. In gradient sparsification, a compressor $Comp_k$ is applied on each worker to locally select $k$, $k \leq d$, gradients for aggregation and $Comp_k \in \{\text{Top}_k, \text{Rand}_k\}$ (Stich et al., 2018). $Comp_k(\boldsymbol{g}_t^p) \in \mathbb{R}^d$ zeros out $(d-k)$ elements of $\boldsymbol{g}_t^p$ and keeps $k$ elements unchanged. The zeroed-out $d-k$ elements are stored as residual $\boldsymbol{\epsilon}_t^p$ for the next iteration. Formally, the model parameters are updated by

$$\boldsymbol{x}_{t+1} = \boldsymbol{x}_t - \eta_t \frac{1}{P} \sum_{p=1}^{P} Comp_k(\boldsymbol{g}_t^p + \boldsymbol{\epsilon}_t^p) \text{ and } \boldsymbol{\epsilon}_{t+1}^p = \boldsymbol{g}_t^p + \boldsymbol{\epsilon}_t^p - Comp_k(\boldsymbol{g}_t^p + \boldsymbol{\epsilon}_t^p), \tag{2}$$

where $\boldsymbol{\epsilon}_t^p \in \mathbb{R}^d$ and $\boldsymbol{\epsilon}_0^p = \boldsymbol{0}$. In theory, distributed SGD with gradient sparsification (e.g., $\text{Top}_k$, $\text{Rand}_k$ and any other $k$-contraction operators) with error compensation has been proved to have the

---

[1]The ring-based AllReduce collective can achieve the bandwidth optimal performance that is not related to the number of workers, but there exist latency terms that will increase with increased number of workers.

same order of convergence rate as vanilla SGD for both convex and non-convex problems if the number of iterations is large (Wangni et al., 2018; Stich et al., 2018; Alistarh et al., 2018; Jiang & Agrawal, 2018; Karimireddy et al., 2019; Tang et al., 2019; Zheng et al., 2019). The convergence rates are derived with a key contraction property of the sparsification operator $Comp_k$ ($Top_k$ or $Rand_k$) (Stich et al., 2018; Alistarh et al., 2018), that is

$$\mathbb{E}_C[\|\boldsymbol{x} - Comp_k(\boldsymbol{x})\|^2] \leq (1 - k/d)\|\boldsymbol{x}\|^2, \forall \boldsymbol{x} \in \mathbb{R}^d, \tag{3}$$

where $\mathbb{E}_C$ is the expectation taking on the compressor and $\|\cdot\|$ is the $\ell_2$-norm. For any $\boldsymbol{x} \in \mathbb{R}^d$, $Top_k(\boldsymbol{x}) \in \mathbb{R}^d$ selects the top $k$ largest elements (in terms of the absolute value) of $\boldsymbol{x}$ with corresponding indices and sets other $d - k$ elements to zeros; while $Rand_k(\boldsymbol{x}) \in \mathbb{R}^d$ randomly (in a uniform distribution) selects $k$ elements from $\boldsymbol{x}$ with corresponding indices and other $d-k$ elements are zeros. It is obvious that

$$\|\boldsymbol{x} - Top_k(\boldsymbol{x})\|^2 \leq \|\boldsymbol{x} - Rand_k(\boldsymbol{x})\|^2 \text{ and } \mathbb{E}_R[\|\boldsymbol{x} - Rand_k(\boldsymbol{x})\|^2] = (1 - k/d)\|\boldsymbol{x}\|^2. \tag{4}$$

Existing studies use the same error estimate for both $Top_k$ and $Rand_k$ in distributed SGD by exploiting the properties of (4), which cannot differentiate the convergence behavior of two operators. In practice, however, TopK-SGD has a much faster convergence speed (in term of iterations) than SGD with $Rand_k$ (RandK-SGD) as empirically shown in (Stich et al., 2018). We also compare the convergence performance between TopK-SGD and RandK-SGD on a 16-worker distributed setting with three popular convolutional neural networks (VGG-16 (Simonyan & Zisserman, 2014), ResNet-20 and ResNet-50 (He et al., 2016)). Our results are shown in Fig. 1. We observe that TopK-SGD achieves very similar performance to the original distributed SGD (Dense-SGD), while RandK-SGD has much slower convergence than TopK-SGD. RandK-SGD even cannot converge on ImageNet. Therefore, though existing studies show that TopK-SGD and RandK-SGD have the same convergence bound, their theoretical results cannot explain the performance gap between TopK-SGD and RandK-SGD. Even some work (Karimireddy et al., 2019; Tang et al., 2019) exploits $\delta \leq 1$ to replace $k/d$ in (3), they also fail to identify exact $\delta$ to distinguish $Top_k$ and $Rand_k$.

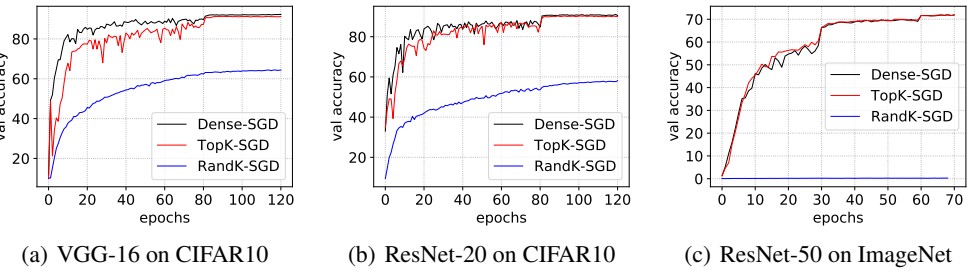

(a) VGG-16 on CIFAR10          (b) ResNet-20 on CIFAR10          (c) ResNet-50 on ImageNet

Figure 1: Convergence comparison between original distributed SGD (Dense-SGD), $Top_k$ sparsification (TopK-SGD) and $Rand_k$ sparsification (RandK-SGD) at 16 distributed workers on the CIFAR10 (Krizhevsky et al., 2010) and ImageNet (Deng et al., 2009) data sets. $k = 0.001d$ for TopK-SGD and RandK-SGD.

In this paper, we dive into the details of the $Top_k$ operator in distributed SGD when training DNNs and provide a tighter bound than inequality (3) to explain the good convergence performance of TopK-SGD. The observation of gradients with $Top_k$ sparsification further enables us to propose a new computational-efficient selection algorithm for gradient which preserves the convergence property. Our contributions are summarized as follows.

**Contributions.** (1) We empirically study the details of local stochastic gradients and observe that the coordinates of gradient follow bell shaped distributions through extensive experiments. (2) The bell shaped distribution enables us to intuitively explain that $Top_k$ should have a much tighter bound than $Rand_k$, and we exploit the distribution property to formulate how $Top_k$ outperforms $Rand_k$. (3) We design and implement an approximate top-$k$ selection algorithm[2], which is much more efficient than existing top-$k$ selection algorithms on GPUs. As compared with the existing sampling-based approximate top-$k$ selection algorithm, we improve the scaling efficiency by 12-50% on our 16-GPU cluster.

---

[2]Our system implementation will be made open-source after the review process.

## 2 RELATED WORK

**Gradient Quantization.** In distributed training of neural networks, the communicated gradients can be quantized to low-bit precision (e.g., 16-bit (Micikevicius et al., 2018; Jia et al., 2018), 3-bit (Wen et al., 2017), 2.8-bit (Alistarh et al., 2017; Karimireddy et al., 2019) and even 1-bit (Seide et al., 2014; Strom, 2015)) while preserving nearly consistent convergence performance with the full precision (32-bit) counterpart. Recently general frameworks of gradient quantization with error compensation are proposed to generalize the theoretical results of low-bit communication (Wu et al., 2018; Jiang & Agrawal, 2018; Karimireddy et al., 2019; Tang et al., 2019; Haddadpour et al., 2019). However, the quantization method can only reduce the communication traffic in $32\times$ (i.e., 1-bit vs. 32-bit), and it could not be enough for large-scale models or low-bandwidth network connections.

**Gradient Sparsification.** Compared to gradient quantization, gradient sparsification is a much more promising communication traffic reduction technique as it can sparsify up to three orders of magnitude gradients be zero with little impact on the model convergence (Strom, 2015; Dryden et al., 2016; Aji & Heafield, 2017; Chen et al., 2018; Lin et al., 2018; Shi et al., 2019a). Due to the much success of gradient sparsification (e.g., Top-$k$ sparsification) in significantly reducing the communication traffic (Lin et al., 2018; Sun et al., 2019), much recent work tries to build theoretical guarantees for SGD with gradient specification (Wangni et al., 2018; Stich et al., 2018; Alistarh et al., 2018; Jiang & Agrawal, 2018; Shi et al., 2019b; Karimireddy et al., 2019; Tang et al., 2019). These theoretical frameworks try to generalize the sparsification operator with the bound of inequality (3) to derive the convergence results for SGD with gradient sparsification. However, the existing analysis fails to go insight into the details of gradient sparsification of $\text{Top}_k$ which could have better convergence than other compression operators (e.g., $\text{Rand}_k$).

**Gradient Distribution[3].** Glorot & Bengio (2010) study the distribution of activation values of DNNs and also their corresponding gradients. They empirically showed that back-propagated gradients have Gaussian-like distributions, which helps understand the difficulty of training deep neural networks. A similar plot is shown in (Micikevicius et al., 2018), where the distribution of gradients helps analyze if the 16-bit representation of gradients would be overflow or underflow. These work has demonstrated that the gradients during training are likely located near zeros. We extend the similar studies on the gradient distribution for TopK-SGD.

## 3 STUDY ON STOCHASTIC GRADIENTS

### 3.1 GRADIENT DISTRIBUTION

In previous gradient sparsification studies (Strom, 2015; Dryden et al., 2016; Aji & Heafield, 2017; Chen et al., 2018; Lin et al., 2018), the basic rule of sparsification is to select "significant" elements of the gradients because they contribute more to the updates. The $\text{Top}_k$ operator selects the exact local top-$k$ elements of gradients so that it achieves nearly consistent convergence performance with Dense-SGD. Therefore, we would like to understand what is the difference between "significant" elements of the gradients and randomly selected ones. We conduct extensive experiments to study the gradient distributions on three areas of deep learning applications, including image classification, language modeling, and speech recognition. The selected models are: 1) **Feed-forward Neural Networks (FNNs)**. An FNN with three hidden fully connected layers (FNN-3) on the MNIST (LeCun, 1998) data set. 2) **Convolutional Neural Networks (CNNs).** LeNet-5 (LeCun et al., 2015) on MNIST, ResNet-20 (He et al., 2016) and VGG-16 (Simonyan & Zisserman, 2014) on CIFAR10 (Krizhevsky et al., 2010). And 3) **Recurrent Neural Networks (RNNs).** Long Short Term Memory networks (LSTMs) on the Penn Treebank (PTB) (Marcus et al., 1993) and the AN4 (Acero, 1990) data sets. For PTB, we adopt a 2-layer LSTM model (LSTM-PTB) with 1500 hidden units per layer, and for AN4, we use a 5-layer LSTM model (LSTM-AN4) with 800 hidden units per layer.

The details of the experimental settings are shown in Table 1. As the compression operator is applied on the gradients, we first measure the distributions of the gradient's elements (histograms) on Dense-SGD. The results demonstrate the similar shapes as (Glorot & Bengio, 2010), while ours covers various applications (refer to Appendix A.2). Our interest is on TopK-SGD to check if

---

[3]The distribution we discussed in this paper is over coordinates on a particular vector (e.g., activation outputs, full gradients).

Table 1: Experimental settings. All models are trained by SGD with a 0.9 momentum. "BS" is the mini-batch size at each worker. "LR" is the initial learning rate which is decayed during training. The hyper-parameters are set to cover various weight initialization methods, activation functions, batch sizes and learning rates with proper convergence performance.

| Type | Model | # Params | Weight Init. | Activation | BS | LR | Data Set |
|------|-------|----------|--------------|------------|-----|------|----------|
| FNN | FNN-3 | 199,210 | Xavier | ReLU | 128 | 0.01 | MNIST |
| CNN | LeNet-5 | 61,706 | Xavier | ReLU | 128 | 0.01 | |
| | ResNet-20 | 269,722 | Xavier, Kaiming | ReLU | 32 | 0.1 | CIFAR10 |
| | VGG-16 | 14,728,266 | Kaiming | ReLU | 128 | 0.1 | |
| RNN | LSTM-PTB | 66,034,000 | Uniform | Tanh | 20 | 22 | PTB |
| | LSTM-AN4 | 27,569,568 | Xavier | Tanh | 4 | 0.0002 | AN4 |

gradients distributions perverse the same properties as Dense-SGD. During the training process of TopK-SGD ($k = 0.001d$ for a $d$-dimension model), we measure the histograms of local gradients accumulated with the residuals (i.e., $\boldsymbol{u}_t^p = \boldsymbol{g}_t^p + \boldsymbol{\epsilon}_t^p$). The histograms of $\boldsymbol{u}_t^1$ with different $t$ on different models are shown in Fig. 2, where we only show the gradients from the first worker as different workers have very close gradient distributions. The corresponding cumulative distributions are presented in Appendix A.1. It is seen that different models have different shapes on the accumu-

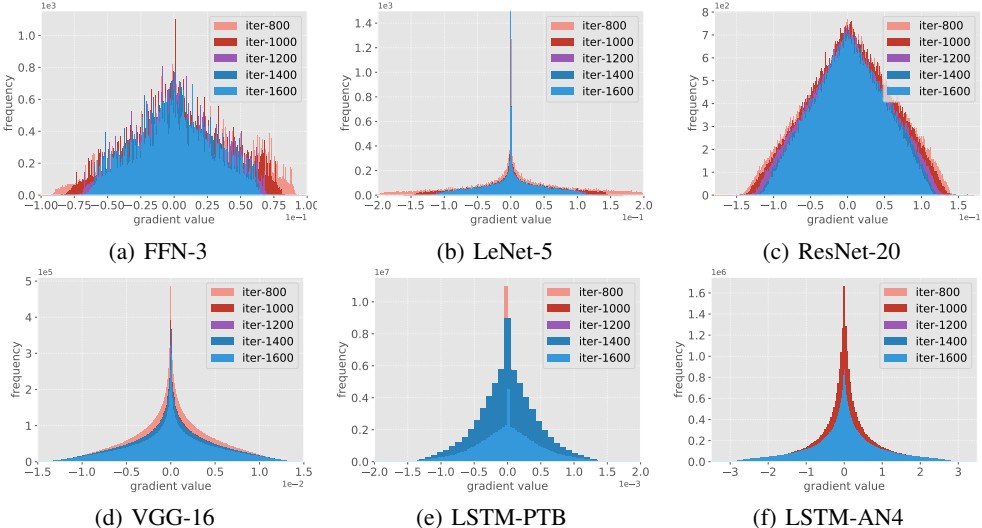

Figure 2: The histograms of $\boldsymbol{u}_t^1$ of TopK-SGD. For each model, the gradient histograms are plotted every 200 iterations from iteration 200 to 1600 (other iterations have similar shapes).

lated gradients, but one common feature is that most coordinates of $\boldsymbol{u}_t$ are close to zero. Compared to the full gradient SGD (Appendix A.2), TopK-SGD shows wider distributions, which could be mainly caused by residual accumulation. When selecting top-$k$ largest values (in terms of absolute values) from $\boldsymbol{u}_t$, the selected values should be located at the left and right sides on the histograms. Therefore, performing $\text{Top}_k$ on $\boldsymbol{u}_t$ should generate a vector whose $\ell_2$-norm is very close to that of $\boldsymbol{u}_t$, that is $\|\text{Top}_k(\boldsymbol{u}_t)\|^2 \lessgtr \|\boldsymbol{u}_t\|^2$. The intuitive result inspires us to formulate how much close of $\|\text{Top}_k(\boldsymbol{u}_t)\|^2$ to $\|\boldsymbol{u}_t\|^2$. Specifically, we would like to derive a variable $\gamma \leq (1 - k/d)$ such that $\|\boldsymbol{u}_t - \text{Top}_k(\boldsymbol{u}_t)\|^2 \leq \gamma \|\boldsymbol{u}_t\|^2$ holds.

## 3.2 THEORETICAL ANALYSIS AND RESULTS

We investigate the $\text{Top}_k$ operator on $\boldsymbol{u}_t^p = \boldsymbol{g}_t^p + \boldsymbol{\epsilon}_t^p$ (for ease of presentation, we use $\boldsymbol{u}$ to denote $\boldsymbol{u}_t^p$).

**Error estimation of $\text{Top}_k$.** Let $\boldsymbol{\pi}$ denote a sorted vector of $|\boldsymbol{u}|/\|\boldsymbol{u}\|_\infty$ in a descending order. That is $\boldsymbol{\pi}_{(i)} \geq \boldsymbol{\pi}_{(i+1)} \geq 0$ for $i = 1, 2, ..., d-1$, where $\boldsymbol{\pi}_{(i)}$ is the $i^{th}$ element of $\boldsymbol{\pi} \in \mathbb{R}^d$. Then we have

$$\frac{\|\boldsymbol{u} - \text{Top}_k(\boldsymbol{u})\|^2}{\|\boldsymbol{u}\|^2} = \frac{\|\boldsymbol{u} - \text{Top}_k(\boldsymbol{u})\|^2/\|\boldsymbol{u}\|_\infty^2}{\|\boldsymbol{u}\|^2/\|\boldsymbol{u}\|_\infty^2} = \frac{\|\tilde{\boldsymbol{u}} - \text{Top}_k(\tilde{\boldsymbol{u}})\|^2}{\|\tilde{\boldsymbol{u}}\|^2} = \frac{\sum_{i=k+1}^d \boldsymbol{\pi}_{(i)}^2}{\sum_{i=1}^d \boldsymbol{\pi}_{(i)}^2}, \quad (5)$$

where $\tilde{u} = u/\|u\|_\infty$. Assume that $u_{(i)}$ follows a bell shaped distribution (e.g., Fig. 3(a)), and $\pi^2$ is a decreasing function w.r.t. $i$ as shown in Fig. 3(b). In order to evaluate Eq. (5), it is essential to calculate the area under the curve of $\pi^2$. As illustrated in Fig. 2, one can empirically prove that $\pi^2$ is convex and it is always less than the reference line ($y = -i/d + 1$) if $u$ follows bell shaped distributions. Considering the areas of $A_1, A_2, A_3,$ and $A_4$ shown in Fig. 3(c), we have

$$\frac{\sum_{i=k+1}^{d} \pi_{(i)}^2}{\sum_{i=1}^{d} \pi_{(i)}^2} = \frac{A_1}{A_1 + A_2 + A_3} \leq \frac{A_1 + A_4}{A_1 + A_2 + A_4}. \tag{6}$$

Due to the space limit, the proof of the inequality is put in Appendix A.4. Then we have

$$\frac{A_1}{A_1 + A_2 + A_3} \leq \frac{A_1 + A_4}{A_1 + A_2 + A_4} = \frac{\text{Area of } MDB}{\text{Area of } OCB} = \frac{\text{Area of } EBD}{\text{Area of } OAB} = \left(1 - \frac{k}{d}\right)^2, \tag{7}$$

where the second equality can be obtained from the similarity of triangle $\triangle MDB \sim \triangle COB$ and $\triangle EDB \sim \triangle AOB$, i.e.,

$$\frac{\text{Area of } MDB}{\text{Area of } OCB} = \frac{MD}{CO} = \frac{DB}{OB} = \frac{ED}{AO} = \frac{\text{Area of } EBD}{\text{Area of } OAB}. \tag{8}$$

Putting altogether, we have

$$\|u - \text{Top}_k(u)\|^2 / \|u\|^2 \leq (1 - k/d)^2 =: \gamma \tag{9}$$

and eventually

$$\|u - \text{Top}_k(u)\|^2 \leq \gamma \|u\|^2 \leq (1 - k/d) \|u\|^2, \tag{10}$$

where $\gamma = (1 - k/d)^2$. The last inequality is always true as $|1 - k/d| \leq 1$. Our results can be

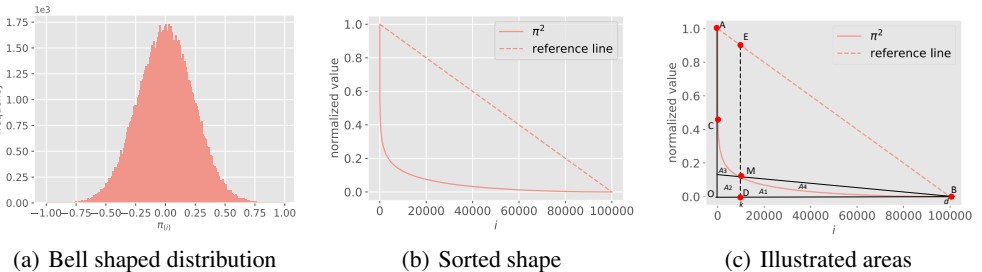

(a) Bell shaped distribution     (b) Sorted shape     (c) Illustrated areas

Figure 3: The shape of $\pi_{(i)}^2$ with different $i$ with $d = 100,000$ and $\sigma = 1$.

summarized as the following theorem.

**Theorem 1.** *Assume that $u \in \mathbb{R}^d$ follows a bell shaped distribution and $\pi^2$ is convex and less than the line $y = -i/d + 1$, then we have*

$$\|u - Top_k(u)\|^2 \leq (1 - k/d)^2 \|u\|^2. \tag{11}$$

*Furthermore, it can be rearranged into the form that*

$$\|u - Top_k(u)\|^2 \leq (1 - \delta) \|u\|^2, \quad \text{where } \delta = (2kd - k^2)/d^2. \tag{12}$$

**Convergence Bound of TopK-SGD.** We use the same assumptions on the objective function $f : \mathbb{R}^d \to \mathbb{R}$ as (Karimireddy et al., 2019). The assumptions are: 1) $f$ is $L$-smooth and 2) $f$ has a moment bound (i.e., $\mathbb{E}[g] = \nabla f(x)$ and $\mathbb{E}[\|g\|^2] \leq G^2$ for some $G > 0$, where $g$ is a stochastic gradient and $x$ is the model parameter). Therefore, we can directly use the the bound formulation of convergence rate with $\delta$ from (Karimireddy et al., 2019) in Remark 4.

**Theorem 2.** *If we set $\eta_t = \frac{1}{\sqrt{T+1}}$ for running TopK-SGD and under the assumptions of $f$, we have*

$$\min_{t \in [T]} \mathbb{E}[\|\nabla f(x_t)\|^2] \leq \frac{4(f(x_0) - f^*) + LG^2}{2\sqrt{T+1}} + \frac{4L^2 G^2 (1 - \delta)}{\delta^2 (T + 1)}, \tag{13}$$

*where $f^*$ is the optimal solution.*

The theorem indicates that after $T \geq O(1/\delta^2)$ iterations, the first term of the right-hand side of inequality (13) will dominate the bound so that the convergence rate becomes $O(1/\sqrt{T})$ which matches the rate of vanilla SGD. Note that our derived bound of $\delta = (2kd - k^2)/d^2$ is much tighter than $k/d$ in previous studies (Stich et al., 2018; Alistarh et al., 2018; Jiang & Agrawal, 2018; Shi et al., 2019b; Karimireddy et al., 2019). Let $c = d/k$ denote the compression ratio of gradients. Previous results ($\delta = 1/c$) indicate that RandK-SGD or TopK-SGD should run after $T \geq O(c^2)$ iterations to make it catch up the convergence rate of Dense-SGD. Using inequality (10) for TopK-SGD, it just requires $T \geq O(c^4/(2c-1)^2)$ iterations to have the full gradient convergence rate. The result gives the explanation to why TopK-SGD can easily achieve nearly consistent convergence performance to Dense-SGD, while RandK-SGD could not (as shown in Fig. 1).

### 3.3 GAUSSIAN$_k$: AN APPROXIMATE TOP$_k$ OPERATOR

Though TopK-SGD has a good convergence property with a significantly reduced communication size in distributed SGD, the exact top-$k$ selection is not friendly to many-core processors like GPUs (Shanbhag et al., 2018). Inefficient Top$_k$ could make the overall wall-clock time worse. For example, training a ResNet-50 (He et al., 2016) model on ImageNet (Deng et al., 2009) on an Nvidia Tesla V100 GPU with a mini-batch size of 128 requires around 0.46 seconds per iteration[4]. When we distribute the training to 16 Tesla V100 GPUs connected with 10 Gbps Ethernet (10GbE), the communication time of full gradients ($d = 25,557,032$) is around 0.2 seconds. However, the Top$_k$ operator with $k = 0.001d$ on ResNet-50 with the Tesla V100 GPU consumes 0.4 seconds. The 0.2-second communication overhead is saved, but it introduces another 0.4 seconds, which makes the training efficiency even worse. In DGC-SGD (Lin et al., 2018), the authors proposed to sample only 0.1% to 1% of the gradients to estimate the threshold hierarchically, which requires to invoke top-$k$ selection twice on the subsets of the original vector. For ease of reference, we use DGC$_k$ to denote the hierarchical sampling method in selecting the largest top-$k$ gradients. In RedSync-SGD (Fang et al., 2019), the authors proposed a trimmed top-k selection algorithm (Trimmed$_k$) to select top gradients for CNNs by heuristically searching the threshold with moving the ratio between the maximum value and the average value. However, Trimmed$_k$ could use a threshold that is much smaller than the exact top-$k$ threshold so that the number of selected gradients is much higher than $k$.

---

**Algorithm 1** Gaussian$_k$

**Input:** Stochastic gradients with residuals $\boldsymbol{u}_t^p$
**Input:** $k$ and dimension $d$
1: Initialize $\hat{\boldsymbol{u}}$ as a zero vector with $d$ dimensions;
2: $\mu, \sigma$ = mean and std of vector $\boldsymbol{u}_t^p$;
3: $p = 1 - k/d$;
4: $thres$ = ppf($\boldsymbol{u}_t^p, p, \mu, \sigma$);
5: **for** $i = 0 \rightarrow 3$ **do**
6:      $masks = |\boldsymbol{u}_t^p| > thres$;
7:      $estimated_k$ = # of True values in $masks$;
8:      **if** $estimated_k < 2k/3$ **then**
9:         $thres = 0.5 \times thres$;
10:      **else if** $estimated_k > 4k/3$ **then**
11:         $thres = 1.5 \times thres$;
12:      **else**
13:         break;
14: $\hat{\boldsymbol{u}}[masks] = \boldsymbol{u}_t^p[masks]$;
15: Return $\hat{\boldsymbol{u}}$;

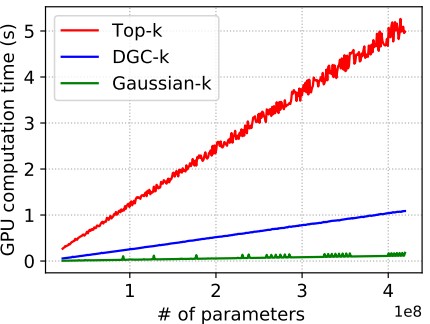

Figure 4: The GPU computation time (lower is better) of Top$_k$, DGC$_k$ and Gaussian$_k$. We use the PyTorch tensor API, "tensor.topk()", for the Top$_k$ operator.

We propose an approximate Top$_k$ operator named Gaussian$_k$ by exploiting the Gaussian-like distribution property of gradients. The key ideas of Gaussian$_k$ are: 1) We regard the $d$-dimensional gradients (i.e., $\boldsymbol{u}_t^p$) at each iteration as a normal distribution with the mean ($\mu$) and standard variance ($\sigma$) which can be directly calculated in an $O(d)$ complexity and the calculations are friendly to GPUs. 2) We estimate the threshold by exploiting the percent point function (ppf) of $\boldsymbol{u}_t^p$ with three parameters: $p = 1 - k/d$, $\mu$ and $\sigma$. 3) As the distribution is not exactly normal, the ppf estimation could result in a threshold that could be slightly smaller or larger than the true threshold. We move

---

[4]The model is trained with the 32-bit floating point without using Tensor Cores of the Tesla V100 GPU.

to the estimated threshold to the left or right side several times such that we can have very close top-$k$ largest absolute values. The algorithm of Gaussian$_k$ is shown in Algorithm 1.

## 4 EXPERIMENTS

As we mainly focus on gradient sparsification, we use the fp32 operations instead of exploiting lower precision for training models. The related software libraries are CUDA-10.1, cuDNN-7.5.0, NCCL-2.3.7, PyTorch-1.1.0, OpenMPI-4.0.1, and Horovod-0.16.4 (Sergeev & Balso, 2018), which are kept the same for all evaluated algorithms.

### 4.1 NUMERICAL RESULTS OF THE TOP$_k$ OPERATOR

To validate the bound of inequality (10), we randomly (in Gaussian distribution) generate a $100,000$ dimension vector and compare the exact value of $\|\boldsymbol{u} - \text{Top}_k(\boldsymbol{u})\|^2 / \|\boldsymbol{u}\|^2$ and $1 - k/d$ with ours derived $(1 - k/d)^2$. We also compare the three bounds in the real-world model training process. The results are shown in Fig. 5. It is seen that both ours and the previous result are in the upper side of the exact value, which indicates the derived bounds hold. With increased $k$, ours becomes better and better than the previous result. However, the exact value is still much lower than ours. The reason is that our bound is derived by the reference line (Fig. 3(b)) but not the original function. Therefore, if the shape of $\boldsymbol{\pi}_{(i)}^2$ can be exactly formulated, one can derive a tighter bound for the Top$_k$ operator than $(1 - k/d)^2$ and we will leave this as our future work.

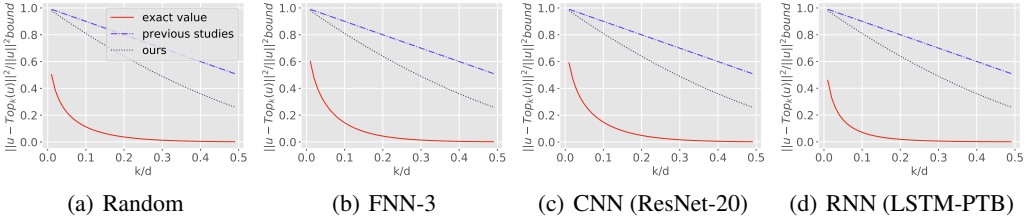

(a) Random     (b) FNN-3     (c) CNN (ResNet-20)     (d) RNN (LSTM-PTB)

Figure 5: The comparison of bounds with a range of $k$.

### 4.2 GPU COMPUTATION EFFICIENCY OF SPARSIFICATION

To evaluate the computing efficiency of different top-$k$ selection algorithms on GPUs, we conduct experiments on an Nvidia Tesla V100 GPU with $d$ ranging from 20 million to 400 million and $k = 0.001d$. The GPU computation speed comparison between Top$_k$, DGC$_k$ and Gaussian$_k$ operators is shown in Fig. 4. For DGC$_k$, we use 1% as suggested in (Lin et al., 2018) to estimate the threshold. Note that tensor operations (e.g., top-$k$ selection, mean and std calculations etc.) are from PyTorch's tensor APIs[5]. The experimental results show that the Top$_k$ operator becomes very slow with a large number of parameters, while Gaussian$_k$ only generates slight overheads. DGC$_k$ also becomes inefficient if $d$ is large. It is crucial for the end-to-end training to have a computing-efficient operator on GPUs such that the extra computation overhead would not limit the system scalability.

### 4.3 CONVERGENCE PERFORMANCE OF GAUSSIANK-SGD.

To demonstrate the convergence performance of GaussianK-SGD, we run 120 epochs on CIFAR10 and 70 epochs on ImageNet with 16 workers. On CIFAR10, the hyper-parameters are listed in Table 1, and on ImageNet, we use a mini-batch size of 32 per GPU and a initial learning rate 0.01. The top-1 validation accuracy of the evaluated models is shown in Fig. 6. Note that for each model, we use the same hyper-parameters for the three SGD algorithms. We can see that our GaussianK-SGD has nearly consistent validation accuracy with TopK-SGD, which indicates that our proposed Gaussian$_k$ operator can select close elements with Top$_k$. The gradient distributions in GaussianK-SGD are similar to TopK-SGD (Appendix A.2). In the evaluated three models, GaussianK-SGD and TopK-SGD have slight accuracy loss (around $0.6\%$-$0.8\%$) compared to Dense-SGD. As suggested in (Lin et al., 2018), the small residuals could have staleness compared to the current gradients

---

[5]https://pytorch.org/docs/stable/tensors.html

so that it could cause the slight accuracy loss. Some optimization tricks in (Lin et al., 2018) like momentum correction would address this problem.

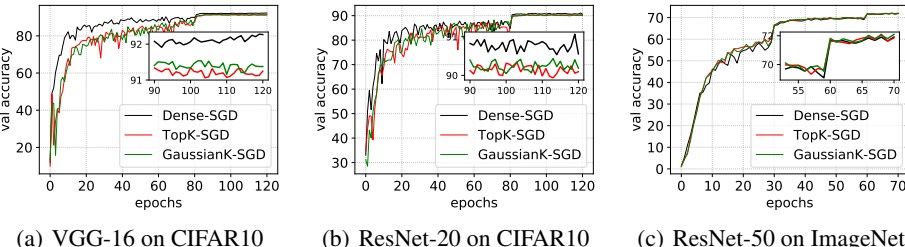

(a) VGG-16 on CIFAR10  (b) ResNet-20 on CIFAR10  (c) ResNet-50 on ImageNet

Figure 6: The convergence performance (top-1 validation accuracy) of distributed SGD with GaussianK-SGD using $k = 0.001d$ compared to TopK-SGD and Dense-SGD on 16 workers.

## 4.4 END-TO-END TRAINING SCALING EFFICIENCY OF GAUSSIANK-SGD.

We evaluate the average iteration time of GaussianK-SGD on the ImageNet (Deng et al., 2009) data set with four popular models (AlexNet (Krizhevsky et al., 2012), VGG-16 (Simonyan & Zisserman, 2014), ResNet-50 (He et al., 2016) and Inception-V4 (Szegedy et al., 2017)) on a 16-GPU cluster compared to Dense-SGD with full gradients, TopK-SGD with the original top-$k$ selection, DGC-SGD (Lin et al., 2018) with hierarchical sampling and RedSync-SGD (Fang et al., 2019) with trimmed top-$k$ selection. The cluster has four nodes connected with 10GbE, and each node contains four Nvidia Tesla V100 GPUs (the PCIe version with 32GB memory). $k = 0.001d$ for all the sparsified algorithms. The results are shown in Table 2, which shows that TopK-SGD and RedSync-SGD are even slower than Dense-SGD on the 16-GPU cluster, while our GaussianK-SGD runs much faster than other algorithms. Specifically, GaussianK-SGD is $1.19\times$-$2.33\times$ faster than Dense-SGD, $1.36\times$-$3.63\times$ faster than TopK-SGD, and $1.11\times$-$1.51\times$ faster than DGC-SGD, respectively. Even on the VGG-16 model, which has several large-size fully connected layers, GaussianK-SGD can achieve 85.5% scaling efficiency on the 16-GPU cluster with low-bandwidth Ethernet.

Table 2: Wall-clock time of end-to-end training with ImageNet on 16 Tesla V100 GPUs. The batch size for each GPU is 128, and the input image resolution is 224×224. Scaling efficiency is defined by $\frac{T_{16}}{16T_1}$, where $T_1$ is the throughput of single GPU training, and $T_{16}$ is the overall system throughput of distributed training on 16 GPUs with weak-scaling.

| Model | Iteration Time (s) | | | | | Scaling Efficiency (%) | | | | |
| --- | --- | --- | --- | --- | --- | --- | --- | --- | --- | --- |
| | Dense | TopK | DGC | RedSync | GaussianK | Dense | TopK | DGC | RedSync | GaussianK |
| AlexNet | 0.571 | 0.891 | 0.369 | 7.203 | **0.245** | 14.1 | 9.0 | 21.8 | 1.11 | **32.8** |
| VGG-16 | 2.068 | 3.010 | 1.540 | 14.670 | **1.311** | 54.2 | 37.2 | 72.8 | 7.6 | **85.5** |
| ResNet-50 | 0.699 | 0.810 | 0.655 | 2.588 | **0.586** | 65.8 | 56.8 | 70.2 | 17.9 | **78.5** |
| Inception-V4 | 1.022 | 1.268 | 0.916 | 3.953 | **0.787** | 67.5 | 54.4 | 75.3 | 17.4 | **87.7** |

## 5 CONCLUSION

In this paper, we first identified that existing theoretical results fail to explain the convergence performance of distributed SGD algorithms with Top-$k$ gradient sparsification (TopK-SGD). Then we empirically studied gradient distributions during training with TopK-SGD through extensive experiments, and observe that the elements of stochastic gradients are mostly located near zero (Gaussian-like distribution). The observation enables us to build a tighter bound for the $\text{Top}_k$ operator based on the empirical assumption of bell shaped distributions of gradients, which makes the convergence property of TopK-SGD explainable. According to the distribution of gradients, we propose an approximate top-$k$ selection algorithm named $\text{Gaussian}_k$ which is much efficient than the existing top-$k$ selection algorithms on GPUs. We finally conduct extensive experiments to verify our derived bound for the $\text{Top}_k$ operator and the convergence performance of distributed SGD with $\text{Gaussian}_k$ (GaussianK-SGD). In terms of the scaling efficiency, GaussianK-SGD achieves up to $2.33\times$, $3.63\times$ and $1.51\times$ faster training speed than full gradient SGD, TopK-SGD and DGC-SGD on a 16-GPU cluster connected with 10 Gbps Ethernet, respectively.

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

# A   APPENDIX

## A.1   CUMULATIVE DISTRIBUTION OF GRADIENTS IN TOPK-SGD

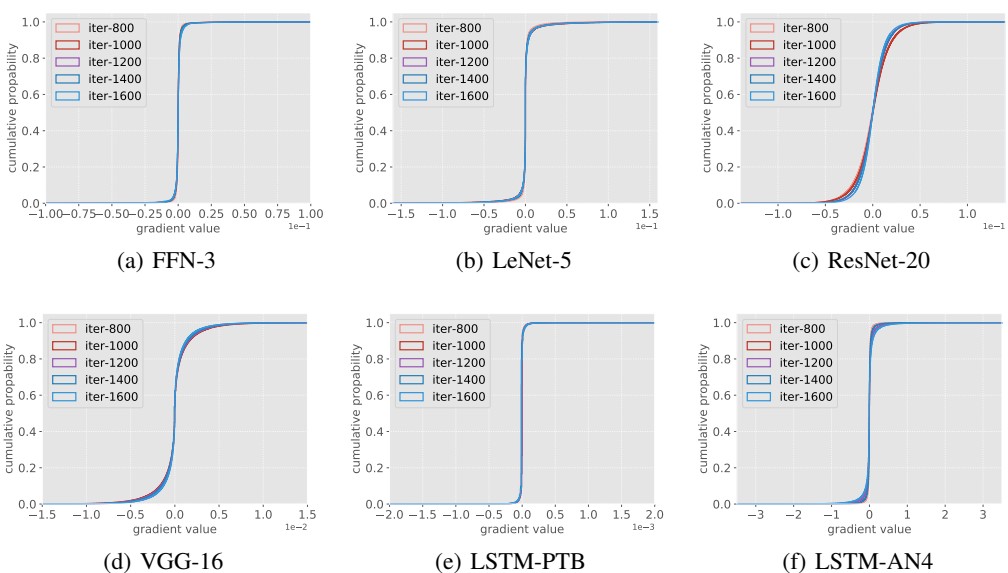

Figure 7: The cumulative distribution of $\boldsymbol{u}_t^1$ during the TopK-SGD training process.

## A.2   GRADIENT DISTRIBUTION ON DENSE-SGD

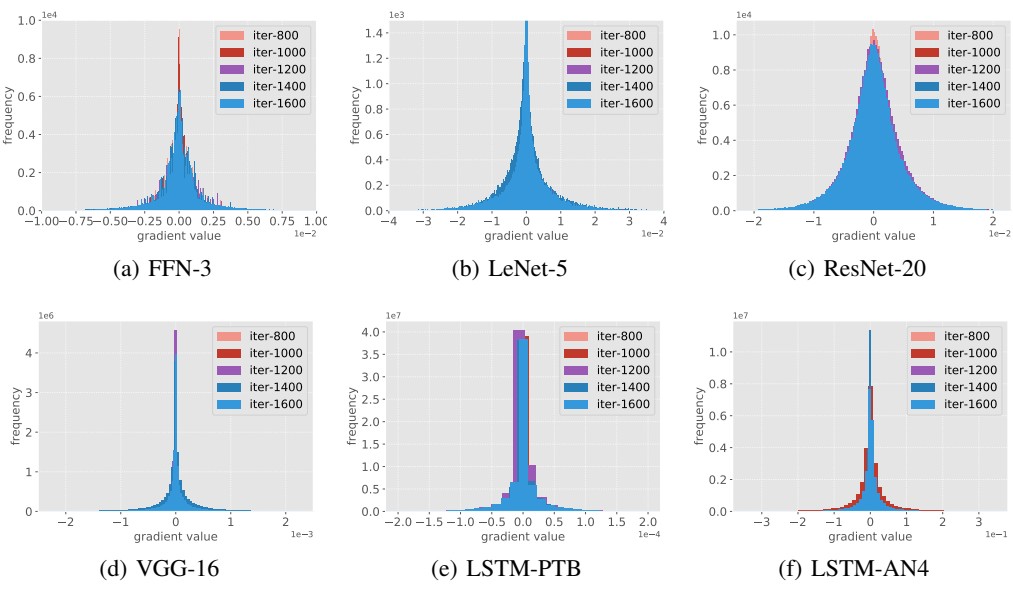

Figure 8: The histograms of $\boldsymbol{u}_t^1$ during the Dense-SGD training process.

### A.3 GRADIENT DISTRIBUTION ON GAUSSIANK-SGD

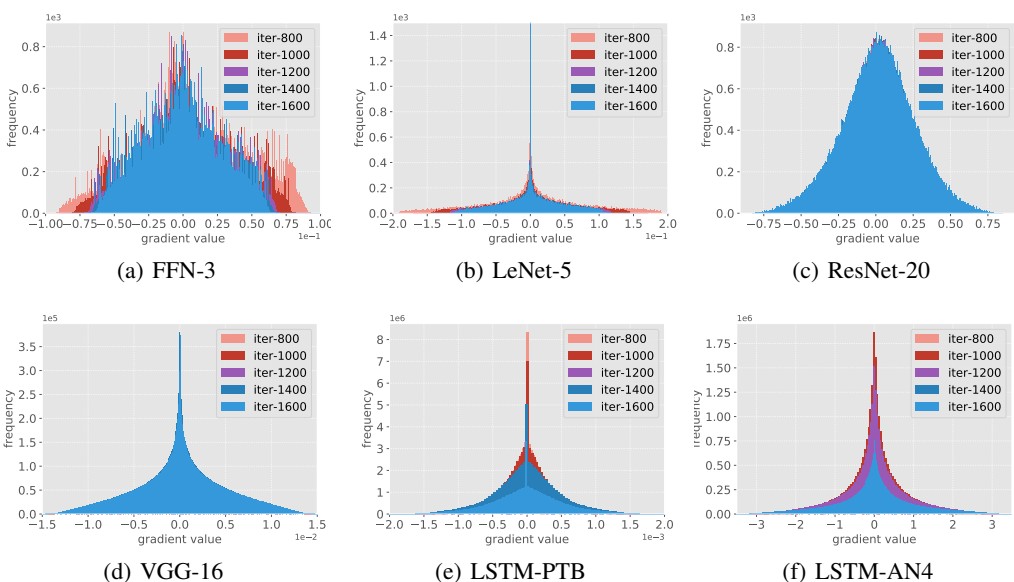

Figure 9: The histograms of $\boldsymbol{u}_t^1$ during the GaussianK-SGD training process.

### A.4 PROOF OF INEQUALITY (6)

$$
\begin{aligned}
&\frac{A_1}{A_1 + A_2 + A_3} \leq \frac{A_1 + A_4}{A_1 + A_2 + A_4} \\
\Leftrightarrow & A_1(A_1 + A_2 + A_4) \leq (A_1 + A_4)(A_1 + A_2 + A_3) \\
\Leftrightarrow & A_1^2 + A_1 A_2 + A_1 A_4 \leq A_1^2 + A_1 A_2 + A_1 A_3 + A_4 A_1 + A_4 A_2 + A_4 A_3 \\
\Leftrightarrow & 0 \leq A_1 A_3 + A_4 A_2 + A_4 A_3.
\end{aligned}
$$

### A.5 SENSITIVITY STUDY OF GAUSSIANK-SGD

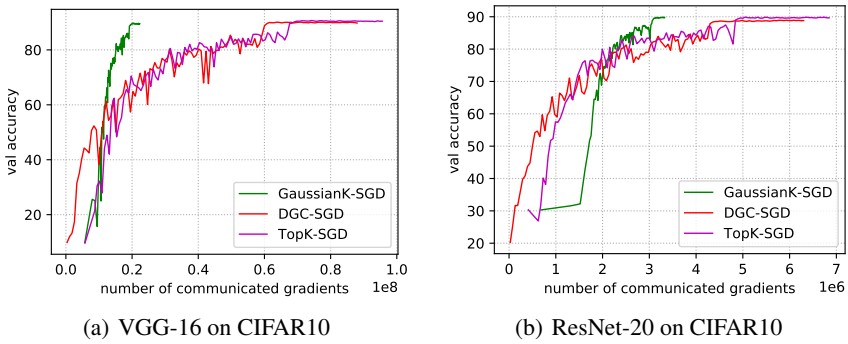

Figure 10: Number of communicated gradients vs. accuracy. $k = 0.001d$

Our proposed Gaussian$_k$ operator could under- or over- sparsify the gradients, which makes the number of selected gradients is larger or smaller than $k$. To demonstrate the sensitivity of GaussianK-SGD to the configured $k$, we first evaluate the accumulated number of communicated gradients over the training process, which is shown in Fig. 10. It is seen that at the first several epochs,

our GaussianK-SGD under-sparsifies the gradients (requires higher communication overheads), and after that, GaussianK-SGD over-sparsifies the gradients (requires lower communication overheads) with little loss of accuracy.

To study the impact of different $k$ on the convergence, we further evaluate the accuracy of GaussianK-SGD by setting $k = 0.01d$ and $k = 0.005d$ on VGG-16 and ResNet-20 models with the same hyper-parameters as Fig. 6. The validation accuracy with different $k$ is shown in Fig. 11. It can be seen that even Gaussian$_k$ would under- or over- sparsify the gradients, GaussianK-SGD performs well on the convergence.

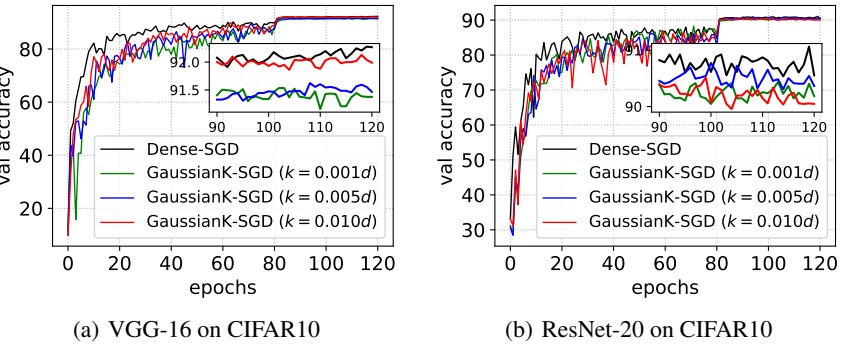

(a) VGG-16 on CIFAR10  (b) ResNet-20 on CIFAR10

Figure 11: Sensitivity of GaussianK-SGD using $k = 0.001d$, $k = 0.005d$ and $k = 0.01d$ compared to Dense-SGD on 16 workers.

