# OpenReview forum: "Understanding Top-k Sparsification in Distributed Deep Learning"
_ICLR.cc/2020/Conference — Reject_

### Official Review · AnonReviewer1 · 2019-10-21
**Official Blind Review #1**

**Rating:** 6

**Review:**

This paper  empirically investigates the distribution of the gradient magnitude in the training of DNNs, based on which a tighter bound is derived over the conventional bound on top-K gradient sparsification.  The authors also propose a so-call GaussianK-SGD to approximate the top-K selection which is shown to be more efficient on GPUs.  Experiments are carried out on various datasets with various network architectures and the results seem to be supportive.  Overall, I find the work interesting and may have real value for communication-efficient distributed training.  On the other hand, I also have following concerns.

1.  The theory part of the work is basically based on empirical observations.  My sense is that it can be more rigorous than its current form.  For instance, the authors may give a concrete form (e.g. Gaussian) of the $\pi$ and derive the area under the curve of  $\pi$ and so on.  Right now, a major part of the derivation is given in a rough sketch.

2. $||\mu||_{\infty}$ should be $||\mu||^{2}_{\infty}$ in Eq. 5.

3. It is not clear to me why the inequality holds in Eq.8.   Could you elaborate on it a bit?  Again, it would be helpful if a concrete function of $\pi$ can be given for derivation.

4. I notice that different distributed training packages are used in experiments, e.g. NCCL and OpenMPI.  It is not clear to me why the experiments are not conducted using a consistent distributed setting.  I don't see authors compare the performance of them.  Also, when comparing the wall-clock time in end-to-end training, are the models trained using the same all-reduce setting?  The authors need to explain. Otherwise, the numbers in Table 2 are not directly comparable.

P.S. rebuttal read.  I will stay with my score.

**Experience Assessment:**

I have read many papers in this area.

**Review Assessment: Checking Correctness Of Derivations And Theory:**

I carefully checked the derivations and theory.

**Review Assessment: Checking Correctness Of Experiments:**

I assessed the sensibility of the experiments.

**Review Assessment: Thoroughness In Paper Reading:**

I read the paper at least twice and used my best judgement in assessing the paper.

---

> ### Author Response · Authors · 2019-11-14
> **Respond to Review #1**
>
> Dear Reviewer, many thanks for your review and comments. We respond to your concerns one by one.
>
> 1.  The theory part of the work ...
> It is non-trivial to have a general form of bell-shaped distribution to derive the exact area under the curve of $\pi^2$ in Fig. 3(c), and thus the exact bound to be derived. Even assuming the concrete from such as Gaussian, it is challenging to calculate the area under the curve of $\pi^2$ formally. So our analysis prefers to use a triangle to derive the bound as shown in Fig. 3(c) in our paper.
>
> 2.  should ...
> We have corrected in the updated version of our paper.
>
> 3. It is not clear...
> We have added the details of the derivation in the updated version of the paper (Appendix A.4). For further work, we would like to calculate the exact area under the curve with some assumptions so that we can have an exact bound.
>
> 4. I notice that ...
> The software packages and the distributed settings are the same for all evaluated algorithms. NCCL and OpenMPI are used together. For Dense-SGD, all-reduce from NCCL API is used for dense gradients aggregation, while for TopK-SGD, DGC-SGD, and GaussianK-SGD, all-gather from NCCL API is used for sparse gradient aggregation.

---

### Official Review · AnonReviewer2 · 2019-10-21
**Official Blind Review #2**

**Rating:** 3

**Review:**

Top-k algorithm is a gradient sparsification method which gains its popularity to train deep networks due to its high compression ratio. However due to its computation overhead, it is not efficient on GPUs. This paper performs empirical study on the distribution of the gradients when training various deep networks, and provides a Gaussian approximation analysis to improve the convergence of the top-k algorithm. Further, the paper proposes to use a Gaussian-k algorithm to perform similar gradient sparsification with a much lower computational cost without losing much convergence accuracy compared to Dense-SGD.

However, the theoretical result seems to me overstated in the sense that it lacks mathematical rigor in the proof and is not clear how much insights it brings to understand better why top-k sparsification algorithms work well in deep learning (Figure 5 shows that the bound is still too tight). It is written after Equation (6) that “One can easily prove that π is convex and it is always less than the reference line (y = −i/d + 1) if u follows bell shaped distributions as illustrated”, however it is not clear me in what sense this is true.  The u are random variables, therefore π is a curve which depends their realizations, hence it is a random curve. Or maybe it holds when d goes to infinity?

The numerical results are specific with k = 0.001d, which makes it hard to see if the Gaussian-k algorithm would still work using different k/d ratio. As shown in Figure 2,  some of the histograms of u^1_t are quite sparse (these plots are hard to read for different iterations, maybe use cdf instead and perform statistical test to check how close to Gaussian distributions), therefore in some of these cases the Gaussian approximation may be poor. It is worth further investigation of the robustness of this algorithm as a replacement of the top-k algorithm.

**Experience Assessment:**

I have published one or two papers in this area.

**Review Assessment: Checking Correctness Of Derivations And Theory:**

I carefully checked the derivations and theory.

**Review Assessment: Checking Correctness Of Experiments:**

I assessed the sensibility of the experiments.

**Review Assessment: Thoroughness In Paper Reading:**

I read the paper at least twice and used my best judgement in assessing the paper.

---

> ### Author Response · Authors · 2019-11-14
> **Respond to Review #2**
>
> Dear Reviewer, thank you very much for your review and comments. Figure 5 shows that our derived bound is much tighter than the conventional one. However, as your concern, it still has a large gap compared to the exact bound. However, it is non-trivial to have a general form of bell-shaped distribution to derive the exact area under the curve of pi^2 in Figure 3(c), and thus the exact bound to be derived. So our analysis prefers to use a triangle to derive the bound as shown in Fig. 3(c) in our paper. Even our derived bound is tighter than the conventional ones, it still has a large gap compared to the exact bound as shown in Fig. 5 in our paper. Our observation and derivation open insight into top-k sparsification and one may derive the exact bound if we know the bell-shaped function. For the convex problem and theorem 1, Actually it is non-trivial to prove the convexity of the curve theoretically, we are sorry that we made this claim without a clear derivation. In this paper, we mainly started with the empirical observation on the gradient distribution, and prove the curve is convex and smaller than the reference line through numerical experiments so that we can derive the tighter bound to explain top-k sparsification. We have corrected the description and make it clear that we empirically prove the curve $\pi^2$ is convex and smaller than the reference line.
>
> For the visualization of the plotted distribution, we have added cdf into our updated version of the paper (Appendix A.1) for better understanding. We did perform normality tests for the distributions, but the results are far from Gaussian, for example, Shapiro-Wilk test gives p-value that is much smaller than 0.01. Actually the distributions are much peakier than Gaussian as shown in Figure 2, so we can only conclude that the distributions are bell-shaped. The bell-shaped distribution enables us to derive the tighter bound of top-k than the conventional one under the empirical assumption on $\pi^2$.
> Actually even on very sparse histograms, our Gaussian-k operator could choose an approximation as we use a 3-step search instead of one step threshold selection using the ppf function. There do exist some cases that Gaussian-k could over- or under- sparsify the gradients, but it preserves the accuracy with low computation overhead and communication overhead, which is demonstrated in Fig. 10 in the updated version. For the robustness, we conduct more experiments to verify the convergence of GaussianK-SGD with different k (e.g., k=0.005d, k=0.01d) in Fig. 11.

---

### Official Review · AnonReviewer3 · 2019-10-24
**Official Blind Review #3**

**Rating:** 3

**Review:**

This paper makes two contributions to gradient sparsification to reduce the communication bottleneck in distributed SGD.
1) Based on an assumption on the distribution of gradient coordinate values that are backed by an empirical study, the paper derives a tighter bound on the approximation quality of top-k gradient sparsification. This result induces better convergence bounds.
2) The authors note that the top-k cannot benefit from the highly parallel architectures popular in ML, and propose an approximate top-k sparsification operator. This operator tries up to three thresholds and checks how many entries are larger than this value. The initial guess is based on approximating the distribution of gradient coordinates with a normal distribution again.

My score is weak reject. I believe that both observations are valid, and that their solutions might be practically meaningful. However, I would like to see the comparison to another baselines [1]. I would also urge the authors to make it more clear that their theoretical results are based on a strong assumption on the distribution of gradients. The top-k approximation algorithm is practical but I find 3-step threshold search inelegant.

Comments:
1) [1] is another baseline -- compare your method with it too.

2) 4.3 Convergence performance "operator can select close elements with Top_k" --- It seems obvious that it can select similar elements. The question is whether the number of elements chosen is accurate. I would like to see this evaluated. It is unclear if this scheme is biased. As far as I can see, it might be over- or under-sparsifying.

3) 3.1 Gradient Distribution "One can easily prove" --- please do so (in the appendix)

4) Theorem 1 - Looks like this can't be true in general. I think it assumes d -> infinity.

5) 3.1 Gradient Distribution "then pi is a decreasing function" --- should this be pi^2. Also in figure 3, the result of Eqn 7 is only correct if the curve if pi^2.

6) Figure 2: I am not convinced that these distributions are 'gaussian'. In fact, they seem peakier. It seems to me that this should improve the results (i.e. make the descending pi^2 curve more convex). If this is true, I would encourage the authors to discuss this. BTW, the distribution is in terms of the whole model, or just one randomly picked layer?

7) Conclusion "theoretically tighter bound" --- because the assumption on the distribution empirical, I find it slightly misleading to call this a 'theoretical bound'. I would urge the authors to make this very clear. (This does not mean I find the bound meaningless)

8) Introduction: "O(d), which generally limits the system scalability" --- The O(d) does not explain scalability in terms of number of workers as is suggested. Note that even though bandwidth scales with O(d) in all reduce, the latency does scale with n. This should not be ignored.

9) Related work/Gradient Sparsification --- Please add a reference for empirical success of top-k. I am not aware of much use outside of academia.

10) Quite a few language errors (some paragraphs/sentences don't make sense at all and there are many cases with missing 'a's etc.)

11) Some of the experimental details are missing
    - what are the learning rates/batch sizes used in experiments.
    - How topK is performed? Layer-wise or for the full gradient.
    - Table 1 "experimental settings": how these values were chosen.
    - Table 2 --- please define how scaling efficiency is computed.
    - Table 2 --- Do these algorithms achieve the same validation accuracy during the training?
    - Figure 1: which k was used in these plots?

12) Figure 6: in VGG-16, the gap of 1 percentage point is quite large. This seems expected as the compression ratio is very high (1000x).

13) Figure 6: Imagenet training scheme is not standard. SOTA validation accuracy for the Imagenet benchmark with Resnet50 is around 76%. Would it also have similar quality loss on later stages as in training VGG or ResNet20 on cifar?

14) Eqn. 8 - I couldn't follow the first inequality.

15) Introdcution: The first few times "distribution of gradients" is mentioned, it was unclear to me if this was over 'coordinates' (as it seems to be), or over 'data points', or 'training time'. Please clarify.


[1] Jiarui Fang, Cho-Jui Hsieh. Accelerating Distributed Deep Learning Training with Gradient Compression.


**Experience Assessment:**

I have read many papers in this area.

**Review Assessment: Checking Correctness Of Derivations And Theory:**

I assessed the sensibility of the derivations and theory.

**Review Assessment: Checking Correctness Of Experiments:**

I assessed the sensibility of the experiments.

**Review Assessment: Thoroughness In Paper Reading:**

I read the paper at least twice and used my best judgement in assessing the paper.

---

> ### Author Response · Authors · 2019-11-14
> **Respond to Review #3 [1/2]**
>
> Dear Reviewer, thank you so much for your detailed and valuable comments. We try to address your concerns one by one.
>
> 1) [1] is ...
> We have added [1] into comparison. Please refer to Table 2 for the performance comparison. There exist several optimization techniques in [1] and it presents two top-k selection algorithms that are related to our work: trimmed top-k selection used for CNNs and threshold binary search selection used for LSTMs. As we evaluated the performance on CNNs, so we compare our Gaussian-k with the trimmed top-k selection algorithm. The trimmed top-k selection algorithm is efficient, but it easily under-sparsifies the gradients such that the number of selected gradients is much larger than k, which makes the communication very slow.
>
> 2) 4.3 Convergence ...
> It is true that it might be over- or under-sparsifying as we use an estimated threshold with the ppf function, but the 3-step search makes the gap between the estimated threshold and the exact threshold smaller. We have evaluated the accumulated number of selected gradients over the training process as in our updated version (Appendix A.5 Fig. 10).
>
> 3) 3.1 Gradient ...
> 4) Theorem 1 ...
> Thank you for pointing out the technical problem of our theorem. Actually it is non-trivial to prove the convexity of the curve theoretically. We are sorry that we made this claim without a clear derivation. In this paper, we mainly started with the empirical observation on the gradient distribution, and prove the curve is convex and smaller than the reference line through numerical experiments so that we can derive the tighter bound to explain top-k sparsification. We have corrected the description and make it clear that we empirically prove the curve pi^2 is convex and smaller than the reference line.  Theorem 1 is also made more clear that it relies on an empirical assumption that $\pi^2$ is convex and smaller than the reference line.
>
> 5) 3.1 Gradient ...
> We have corrected the typos.
>
> 6) Figure 2...
> Yes. These distributions are not Gaussian. We can only clarify the distributions are bell-shaped instead of Gaussian. It can be empirically observed that $\pi^2$ can be more convex. However, it is non-trivial to have a general form of the bell-shaped distribution to derive the exact area under the curve of $\pi^2$. So our analysis prefers to use a triangle to derive the bound as shown in Fig. 3(c) in our paper. We have these discussions in Section 4.1, in which we can see that even our derived bound is tighter than the conventional ones, it still has a large gap compared to the exact bound as shown in Fig. 5 in our paper. Our observation and derivation open insight into top-k sparsification and one may derive the exact bound if we know the bell-shaped function. The distribution is in terms of the whole model.
>
> 7) Conclusion ...
> We have clarified the description as your suggestion.

---

> > ### Author Response · Authors · 2019-11-14
> > **Respond to Review #3 [2/2]**
> >
> > 8) Introduction ...
> > You are correct. We make it more accurate in the description.
> >
> > 9) Related work...
> > Due to the inefficiency of exact top-k selection on GPUs, it’s rare to see the cases that successfully apply exact top-k sparsification in practice. However, there do exist empirical success using a similar idea of gradient sparsification by SenseTime in [1]. The proposed gradient sparsification algorithm is called coarse-grained sparse communication (CSC) in [2], which improves the scalability on dense-GPU clusters. We have added the reference to the updated version of our paper.
> >
> > 10) Quite a few...
> > We have done some proofreadings to correct language errors.
> >
> > 11) Some ...
> > -Batch size (per GPU) and learning rate are 128 and 0.01 for VGG-16 , 32 and 0.01 for ResNet-20, and 32 and 0.1 for ResNet-50.
> > -The Top-k operator is performed for the full gradient.
> > -The hyper-parameters in Table 1 are set to cover various weight initialization methods, activation functions, batch sizes and learning rates with proper convergence performance.
> > -Scaling efficiency is defined by $\frac{T_{16}}{16T_1}$, where $T_1$ is the throughput of single GPU training, and $T_{16}$ is the overall system throughput of distributed training on 16 GPUs with weak-scaling.
> > -As we have evaluated the convergence in Fig. 6, we only evaluate the average iteration time in a fair setting for all evaluated algorithms on Table 2.
> > -k=0.001d in Fig. 1.
> >
> > 12) Figure 1...
> > The compression ratio is one of the factors could affect the model accuracy, which have also been studied in [1][3]. Our paper is mainly to study the bound of top-k sparsification and to propose an approximate top-k selection algorithm (i.e., Gaussian-k), so we mainly to verify if GaussianK-SGD would have the similar convergence performance with the original TopK-SGD and Dense-SGD. The accuracy loss problem for TopK-SGD (and also GaussianK-SGD) can be addressed by the momentum correction technique as suggested in [1][3].
> >
> > 13) Figure 6...
> > In our settings of ImageNet, Dense-SGD only achieves around 71% top-1 validation accuracy, and both TopK-SGD and GaussianK-SGD have similar results with ResNet-50. The main reason why the accuracy is not SOTA is that we resized all JPEG images to 256*256 of the ImageNet data set and then we store the preprocessed JPEG images into an HDF5 file for fast data reading. During training, the 3*256*256 sized images are read out from the HDF5 file and they are further randomly cropped as 3*224*224 for feed-forward computation. Therefore, compared to the cropped images from the original JPEG files, we have some information loss in our HDF5 file, which mainly results in the validation accuracy is lower than SOTA.
> > In fact, there are lots of hyper-parameter settings affecting the validation accuracy (e.g., batch size, learning rate, data augmentation etc.). Our purpose is to provide a fair comparison for all evaluated algorithms (i.e., Dense-SGD, TopK-SGD, and GaussianK-SGD) without tuning the hyper-parameters, i.e., all algorithms are using the same hyper-parameters to train the models. And the hyper-parameters used can have a proper convergence performance even it is not SOTA.
> >
> > 14) Eqn. 8 ...
> > We have added the details of the derivation in Appendix A.4 in the updated version of the paper.
> >
> > 15) Introduction ...
> > Yes, it is over the coordinates of the gradient vector. We have updated it to the paper.
> >
> > [1] Jiarui Fang, Cho-Jui Hsieh. “Accelerating Distributed Deep Learning Training with Gradient Compression.” 2019
> > [2] Sun, Peng, et al. "Optimizing Network Performance for Distributed DNN Training on GPU Clusters: ImageNet/AlexNet Training in 1.5 Minutes." arXiv, 2019.
> > [3] Yujun Lin et al., “Deep Gradient Compression: Reducing the Communication Bandwidth for Distributed Training”.  ICLR, 2018.

---

### Decision · Program_Chairs · 2019-12-19

**Decision:**

Reject

**Comment:**

This paper investigates gradient sparsification using top-k for distributed training. Starting with empirical studies, the authors propose a distribution for the gradient values, which is used to derive bounds on the top-k sparsification. The top-k approach is further improved using a procedure that is easier to parallelize.

The reviewers and AC agree that the problem studied is timely and interesting. However, this manuscript also received quite divergent reviews, resulting from differences in opinion about the rigor and novelty of the results, and perhaps issues with unstated assumptions. In reviews and discussion, the reviewers also noted issues with clarity of the presentation, some of which were corrected after rebuttal. In the opinion of the AC, the manuscript is not appropriate for publication in its current state.